# Microdiversity of the vaginal microbiome is associated with preterm birth

Jingqiu Liao [1,2,9] ✉, Liat Shenhav[3,9], Julia A. Urban [1], Myrna Serrano [4,5], Bin Zhu[4,5], Gregory A. Buck [4,5,6] & Tal Korem [1,7,8] ✉

Preterm birth (PTB) is the leading cause of neonatal morbidity and mortality. The vaginal microbiome has been associated with PTB, yet the mechanisms underlying this association are not fully understood. Understanding microbial genetic adaptations to selective pressures, especially those related to the host, may yield insights into these associations. Here, we analyze metagenomic data from 705 vaginal samples collected during pregnancy from 40 women who delivered preterm spontaneously and 135 term controls from the Multi-Omic Microbiome Study-Pregnancy Initiative. We find that the vaginal microbiome of pregnancies that ended preterm exhibited unique genetic profiles. It was more genetically diverse at the species level, a result which we validate in an additional cohort, and harbored a higher richness and diversity of anti-microbial resistance genes, likely promoted by transduction. Interestingly, we find that *Gardnerella* species drove this higher genetic diversity, particularly during the first half of the pregnancy. We further present evidence that *Gardnerella* spp. underwent more frequent recombination and stronger purifying selection in genes involved in lipid metabolism. Overall, our population genetics analyses reveal associations between the vaginal microbiome and PTB and suggest that evolutionary processes acting on vaginal microbes may play a role in adverse pregnancy outcomes such as PTB.

Preterm birth (PTB), childbirth at <37 weeks of gestation, is the leading cause of neonatal morbidity and mortality[1]. Each year, approximately 15 million infants are born preterm globally, over 500,000 of them in the US[2]. Preterm infants are at a high risk of respiratory, gastrointestinal and neurodevelopmental complications[3]. While a number of maternal, fetal, and environmental factors have been associated with PTB[1,4,5], its etiopathology remains largely unknown, and early diagnosis and effective therapeutics are still lacking.

Over the past decades, growing evidence has pointed to potential involvement of the vaginal microbiome in PTB[6–10]. This involvement has so far been mostly characterized as an ecological process, meaning changes in microbial abundances and vaginal community states. For instance, increased richness and diversity of microbial communities and the presence of particular community state types (CST), have been repeatedly associated with PTB[6,9,11–15]. In addition, vaginal microbiomes of women who delivered preterm appear to be less stable during pregnancy, with some studies reporting a significant decrease in the

[1]Program for Mathematical Genomics, Department of Systems Biology, Columbia University Irving Medical Center, New York, NY, USA. [2]Department of Civil and Environmental Engineering, Virginia Tech, Blacksburg, VA, USA. [3]Center for Studies in Physics and Biology, Rockefeller University, New York, NY, USA. [4]Department of Microbiology and Immunology, School of Medicine, Virginia Commonwealth University, Richmond, VA, USA. [5]Center for Microbiome Engineering and Data Analysis, Virginia Commonwealth University, Richmond, VA, USA. [6]Department of Computer Science, School of Engineering, Virginia Commonwealth University, Richmond, VA, USA. [7]Department of Obstetrics and Gynecology, Columbia University Irving Medical Center, New York, NY, USA. [8]CIFAR Azrieli Global Scholars program, CIFAR, Toronto, ON, Canada. [9]These authors contributed equally: Jingqiu Liao, Liat Shenhav. ✉e-mail: liaoj@vt.edu; tal.korem@columbia.edu

richness and diversity of these microbial communities during pregnancy[6,12].

Multiple endogenous factors, such as hormonal changes, nutrient availability and microbial interactions, and exogenous factors, such as genital infections, antibiotic treatment and exposure to xenobiotics, could trigger ecological processes and alter the vaginal microbial composition[16,17]. These factors may also act as selective pressures that affect genetic variation in the microbial populations that make the vaginal microbiome. Such adaptive evolution in the vaginal environment, even during pregnancy, is highly plausible given the high mutation rates, short generation times, and large population sizes of microbes[18]. They are further supported by observations of rapid adaptation to environmental changes in other human-associated microbial ecosystems[19–21]. The way by which vaginal microbes respond to various selective pressures may, in turn, affect the host, including pregnancy outcomes. Therefore, a comprehensive investigation of the genetic diversity of the vaginal microbiome at the population level, which we term "microdiversity", and the underlying evolutionary forces that shape it, holds promise for a better understanding of the etiopathology of PTB.

Here, we performed an in-depth population genetics analysis and characterized the population structure of the vaginal microbiome along pregnancy and in the context of preterm birth. We used metagenomic sequencing data from 705 vaginal samples collected longitudinally during pregnancy as part of the Multi-Omic Microbiome Study-Pregnancy Initiative (MOMS-PI[6]). Our analyses include samples from 40 women who subsequently experienced spontaneous preterm birth (sPTB) and 135 women who had a term birth (TB). We show that the vaginal microbiome of pregnancies that ended preterm exhibits higher nucleotide diversity at the species level and higher antimicrobial resistance potential. We find that this higher nucleotide diversity is driven by *Gardnerella* spp., a group of central vaginal pathobionts, especially during the first half of pregnancy, and suggest that this may be related to optimization of growth rates in this taxon. We further identify a strong association between evolutionary signatures and sPTB in *Gardnerella* spp., including more frequent homologous recombination and stronger purifying selection. Overall, our results reveal strong associations between the vaginal microbiome and sPTB at the population genetics level, and suggest that evolutionary processes acting on the vaginal microbiome may play a critical role in sPTB, and potentially also in other adverse pregnancy outcomes.

## Results

### The phylogenetic composition of the vaginal microbiome associates with sPTB

We assembled a total of 1078 metagenome-assembled genomes (MAGs) with at least medium quality[22] (>50% completeness, <10% contamination; Supplementary Data 1; Methods) from previously published[6] metagenomic reads generated from 705 vaginal samples[6]. These samples were collected from 175 women visiting maternity clinics in Virginia and Washington at various time points along pregnancy[6], with an average of 3.36 and 3.21 samples for women delivering preterm and at term, respectively (Supplementary Table 1). We clustered these MAGs into 157 species-level phylogroups at the level of 95% average nucleotide identity (ANI), which roughly corresponds to the species level[23]; and selected the most complete MAG with the least contamination as the representative for each phylogroup. These representative MAGs were 86 ± 14% (mean ± SD) complete and 1.1 ± 1.8% contaminated, with 93 (59% of 157) of them estimated to have high quality[22] (>90% completeness and <5% contamination; Supplementary Data 1). Taxonomic assignment of these representative MAGs (Methods) revealed that the phylogroups represent at least 8 phyla, with genome size (adjusted by completeness) ranging from 0.6 to 7.4 Mbps and GC content ranging from 25.3%

to 69.7% (Fig. 1a, Supplementary Data 1). *Actinobacteria* had the most phylogroups detected in the samples, followed by *Firmicutes* and *Bacteroidetes* (Fig. 1a).

Of note, 12 of these species-level phylogroups (PG042-PG053) were assigned to *Gardnerella vaginalis* according to CheckM[24], supporting the existence of multiple genotypes at the species level within the 'species' *G. vaginalis*[25,26]. To better resolve the classification of these *G. vaginalis* phylogroups, we compared the average nucleotide identity (ANI) for the representative MAGs of these phylogroups against updated reference genomes for *Gardnerella*, including *G. vaginalis*, *G. piotti*, *G. leopoldii*, *G. swidsinskii*, and nine species remained to be characterized (gs2-3 and gs7-13)[25]; gs-2-3 and gs7-13 correspond to group 2-3 and 7-13 shown in Fig. 1 in ref. 25. The ANI analysis shows that PG043 represents *G. vaginalis*, PG044 represents *G. swidsinskii*, PG042 represents *G. piotti*, and PG046, PG049, PG051 and PG053 represent *G. gs7*, *G. gs8*, *G. gs13* and *G. gs12*, respectively (Supplementary Fig. 1). The remaining phylogroups (PG045, PG047, PG048, PG050, and PG052) do not cluster with any reference species and may represent previously undocumented species of *Gardnerella*. Here, we refer to phylogroups PG042-PG053 as *Gardnerella* spp.

To understand if the temporal dynamics of the vaginal microbiome is associated with sPTB, we employed compositional tensor factorization (CTF)[27] to assess temporal changes to the composition of the microbiome during pregnancy. This analysis shows a significant separation of women by pregnancy outcomes (PERMANOVA F = 8.0492; p = 0.002; Fig. 1b-e) based on the dynamics of their microbiome composition over time (Fig. 1e), and specifically observed for component 1 in the CTF analysis (Mann-Whitney p = 0.015; Fig. 1c). We further found that the top features contributing to this difference belong to *Lactobacillus helveticus* (PG081), *Lactobacillus crispatus* (PG080), *Lactobacillus gasseri* (PG079), and *Lactobacillus jensenii* (PG076 and PG077) that are associated with TB and *Megasphaera genomosp.* (PG061), *Gardnerella* spp. (PG047, PG050, PG052), and *Atopobium vaginae* (PG041) that are associated with PTB (Fig. 1d); these species were previously found to be associated with pregnancy outcomes[6,9,28]. These results suggest that the vaginal microbiome has a different temporal trajectory during pregnancies ending preterm, consistent with previous findings[6,7], and with *Gardnerella* as an important factor. Overall, our results demonstrate that de-novo metagenomic analysis replicates and expands previous findings with respect to associations between the composition of the vaginal microbiome and sPTB.

Next, we sought to examine the diversity of microbial strains detected within species, and its association with sPTB. We performed the analysis on all phylogroups and found that the strains of *M. genomosp.* showed significantly higher ANI between women who delivered preterm, compared to a null distribution calculated based on ANIs from any two randomly selected women (Permutation p = 0.002, adjusted P < 0.05; Methods), a relationship not observed between women who delivered at term (p = 0.208; Supplementary Fig. 2). This result indicates that *M. genomosp.* were more closely related than expected by chance across women who delivered preterm. It suggests that sPTB-associated vaginal conditions across women may be more conserved, harboring a group of significantly closely related *M. genomosp.* strains, compared to TB-associated vaginal conditions.

### The microdiversity of the vaginal microbiome is higher in the first half of pregnancies that ended preterm, driven by *Gardnerella* species

Human microbes can adapt to host-induced environmental changes (e.g., diet, antibiotics) through genetic variations[29]. Therefore, the microbial populations of the same species in different hosts can have a different genetic structure, which provides them a competitive advantage. These genetic differences, in turn, may be related to the phenotype of the host. To understand the genetic structure of

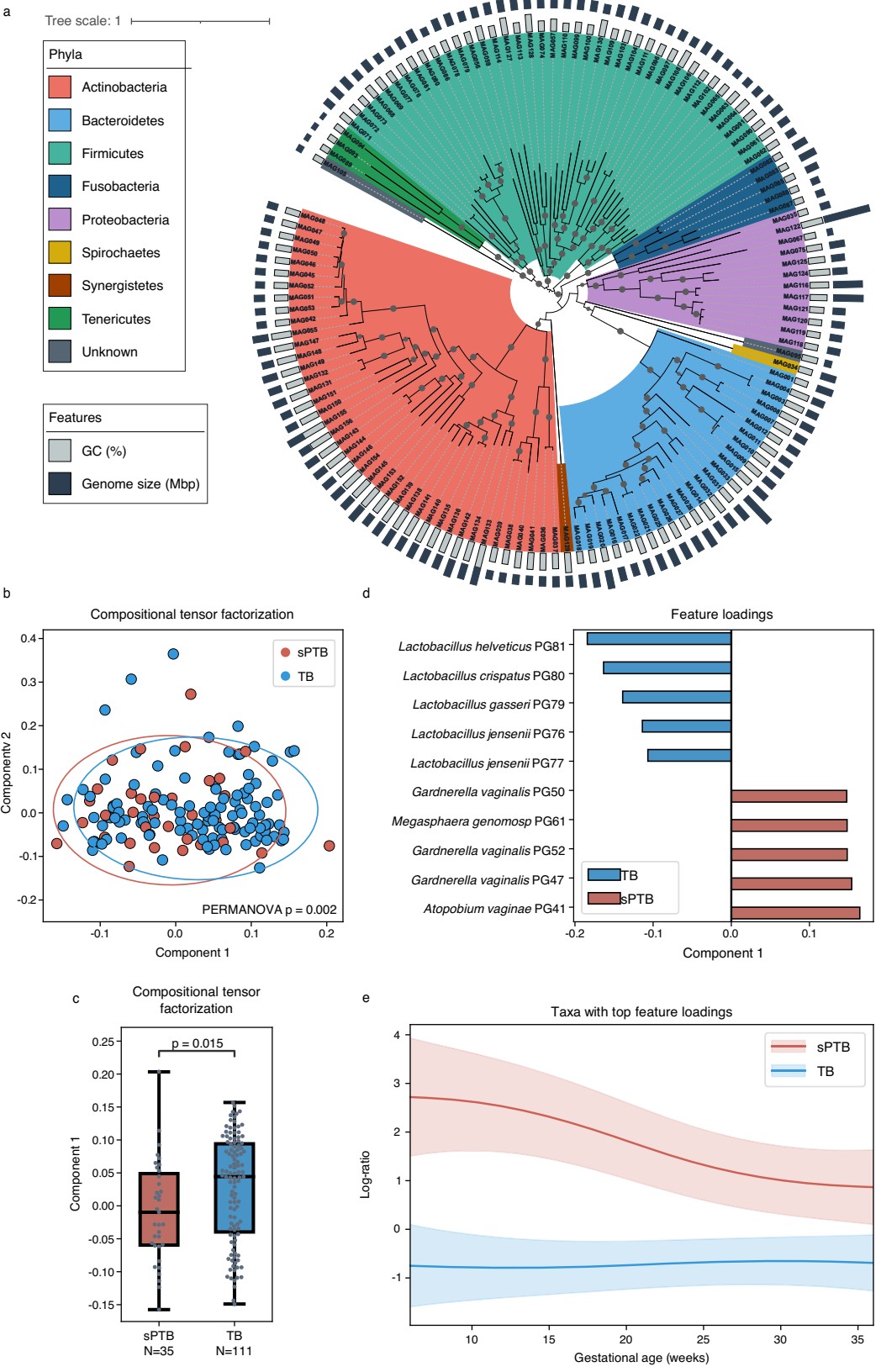

microbial populations in the vaginal environment and its association with pregnancy outcomes, we calculated the nucleotide diversity for each identified phylogroup. Overall, vaginal microbial populations had a significantly higher genome-wide nucleotide diversity in sPTB than in TB (median along pregnancy; Mann-Whitney $p = 0.0073$; Fig. 2a). Stratifying by phylogroups, we found that this difference was mainly

driven by *Gardnerella* spp. ($p = 0.017$; Fig. 2b). *G. piotti* (PG042), *G. swidsinskii* (PG044), and *G.* gs13 (PG051) and a potentially new *Gardnerella* spp. (PG045), along with a phylogroup of *Atopobium vaginae*, a suspected vaginal pathobiont[30], showed significantly higher genome-wide nucleotide diversity in sPTB ($P < 0.05$, adjusted $P < 0.1$ for all; Supplementary Fig. 3a). These results imply that microbial populations

**Fig. 1 | The composition of the vaginal microbiome associated with sPTB.**
**a** Phylogenetic tree of non-redundant MAGs representing 132 species-level phylogroups differing by at least 95% average nucleotide identity (ANI) based on concatenated amino acid (AA) sequences of 120 marker genes. Representative MAGs of 25 phylogroups had <60% of marker genes AA sequence identified and were not included in the tree. Gray nodes indicate a bootstrap value >80. The tree is rooted by midpoint and annotated by the GC content and genome size of the representative MAGs. **b** Compositional tensor factorization (CTF) analysis showing microbiome composition trajectories over gestational ages separated by pregnancy outcomes using the top two ordination axes (Component 1 and Component 2). Each dot represents a subject. **c** Component 1 in the CTF analysis compared between sPTB and TB. Box, IQR; line, median; whiskers, 1.5*IQR; *p*, two-sided Mann-Whitney. **d** Feature rankings of phylogroups colored by preterm birth (sPTB) and term birth (TB) based on Component 1 in the CTF analysis. **e** Log ratio of top and bottom taxa from (**d**) over time, separated to PTB and TB samples. Line, mean; shaded area, 95% CI.

composed of more diverse strains from the same species, and particularly *Gardnerella* spp., are growing in the vaginal environment associated with sPTB.

To understand how the nucleotide diversity of *Gardnerella* spp. changes over time during pregnancy, we analyzed temporal trajectories of term and preterm pregnancies. To this end, we pooled the data from all women in each group, binned pregnancy weeks and used splines to smooth the temporal curves (Methods). We found a significant difference between the temporal trajectories of *Gardnerella* spp. nucleotide diversity in pregnancies ending at term and preterm (Permutation test <0.001 (ref. 31), Wilcoxon signed-rank *P* < 0.003; Methods; Fig. 2c). Specifically, we found that the nucleotide diversity of *Gardnerella* spp. increased at the beginning of pregnancies which ended preterm, with a peak at around gestational week 13, and then dropped to its initial value at around gestational week 20 (Fig. 2c). In comparison, nucleotide diversity of *Gardnerella* spp. in TB remained relatively stable (Fig. 2c). Given that gestational week 20 is the middle of a full-term pregnancy, we subsequently analyzed samples with respect to two time periods - first half (0–19 gestational week) and second half of pregnancy (20–37 gestational week; 37 was chosen to ensure a similar time range for both sPTB and TB). As expected, the nucleotide diversity of *Gardnerella* spp.in sPTB was also significantly higher in sPTB in the first half of pregnancy (median along first half; Mann-Whitney *U* *p* = 0.0091; Fig. 2d), but not in the second half (*p* = 0.71; Fig. 2e). We further found that nucleotide diversity had a significantly stronger correlation with synonymous mutations than with nonsynonymous mutations across *Gardnerella* spp. (paired *t*-test *p* = 0.0011; Supplementary Fig. 3b), suggesting a more important role of purifying selection in shaping genomic diversity. Overall, these results suggest that genetic diversity of *Gardnerella* spp. in the first half of pregnancy is important to birth outcomes, and could perhaps be used as a biomarker for early diagnosis of sPTB.

Analyzing microdiversity across all phylogroups in the two halves of pregnancy, we again found that it was significantly higher in sPTB in the first half of pregnancy (median along first half; Mann-Whitney *U* *p* = 0.0036; Fig. 2f), but not in the second half (*p* = 0.16; Fig. 2g). Notably, we were able to replicate this analysis using data from an additional dataset of vaginal metagenomic sequencing from ten pregnant individuals[32] (median along first and second half; one-sided Mann-Whitney *U* *p* = 0.021 and *p* = 0.12, respectively; Fig. 2h, i). Finally, clinical interventions for risk of preterm birth (e.g., receiving cerclage or progesterone) may alter the microbiome composition, and thus confound our findings. To examine this potential bias, we repeated our analysis on 161 woman who received neither progesterone nor cerclage. Once more, we found that it was significantly higher in sPTB in the first half of pregnancy (*p* = 0.028; Supplementary Fig. 4a), but not in the second half (*p* = 0.12; Supplementary Fig. 4b). Overall, our results demonstrate an increased nucleotide diversity in the vaginal microbiome during pregnancies that ended preterm, in a way that replicates across studies and is not biased by common clinical interventions for prevention of preterm birth.

To understand if any particular genes are driving the association between sPTB and the microdiversity of *Gardnerella* spp. in the first half of pregnancy, we further analyzed nucleotide diversity at the gene level for these species. We identified 21 and 47 genes (out of 825 and 531) in *G. swidsinskii* (PG044) and *G. vaginalis* (PG043), respectively,

that showed significantly different nucleotide diversity between sPTB and TB (median along the first half of pregnancy; Mann-Whitney *P* < 0.05, adjusted *P* < 0.1 for all). These genes included one gene encoding the putative tail-component of bacteriophage HK97-gp10 (*p* = 0.0012) and one gene encoding putative AbiEii toxin, Type IV toxin–antitoxin system (*p* = 5 × 10⁻⁴), which might be involved in the interaction with maternal health[33,34]. To further identify what functions were related to these associations, we then performed functional enrichment analysis (Methods) using the eggNOG functional annotation of genes (Supplementary Data 2). We found that the KEGG pathway 'drug metabolism - other enzymes' (ko00983) was significantly enriched among genes from *G. swidsinskii* (PG044) that had significantly higher microdiversity (*P* < 0.05, adjusted *P* < 0.1; Fig. 2j). This result suggests that the more diverse gene pool in *G. swidsinskii* (PG044) detected in sPTB may be associated with adaptation to drugs present in the environment. This may be consistent with our recent finding that xenobiotics detected in the vaginal environment are strongly associated with sPTB[35].

To verify that the higher nucleotide diversity we observed in sPTB pregnancies was not caused by sampling or sequencing bias, we compared the read count and quality of MAGs obtained from sPTB and TB samples. If this higher diversity is the result of a higher read count in sPTB samples or more complete MAGs, we would expect read count and MAG completeness to be higher in sPTB samples. Instead, we found the completeness and contamination of MAGs assembled from sPTB samples were not significantly different from TB (Mann-Whitney *p* = 0.71 and 0.73, respectively; Supplementary Fig. 5a, b). Next, we assessed the correlation between the number of reads mapped to each phylogroup and its genome-wide diversity. If a higher diversity is caused by more reads mapped to the MAG representing the phylogroup, we would expect a positive correlation between these two measurements. However, only 3 phylogroups (PG064, a *Dialister* spp.; PG102, a *Peptoniphilus* spp.; and PG122, a *Bradyrhizobium* spp., which was likely either a contaminant or misidentified) had a significant positive correlation between read counts and nucleotide diversity (Spearman ρ = 0.70, 0.54, and 0.42, respectively; non-adjusted *p* = 0.035, 0.024, and 0.00015, respectively). In 98% of phylogroups, we did not observe a statistically significant positive correlation (median [IQR] Spearman correlation of −0.067 [−0.19, −0.13]). None of the four *Gardnerella* spp. Phylogroups that showed significantly higher nucleotide diversity in sPTB pregnancies in Supplementary Fig. 3 were significantly positively correlated (Spearman ρ = −0.00, −0.12, −0.02, and −0.35 and p = 0.96, 0.091, 0.77, and 0.00045 for PG042, PG044, PG045, and PG051, respectively; Supplementary Fig. 5c).

Finally, as we have observed a significantly higher read count in sPTB samples (Mann-Whitney *p* = 0.0004 and 0.061 for samples and subjects, respectively; Supplementary Fig. 5d, e, respectively), we subsampled an identical number of reads (10⁵) from each sample, retaining 75% of samples, and repeated our analyses of nucleotide diversity. As with the first analysis (Fig. 2), nucleotide diversity was significantly higher in sPTB pregnancies across all phylogroups, and particularly in *Gardnerella* spp.(Mann-Whitney *p* = 0.015 and *p* = 0.0043, respectively; Supplementary Fig. 5f, g, respectively). Similarly, in the first half of pregnancy, the nucleotide diversity of *Gardnerella* spp.was significantly higher in sPTB (*p* = 0.026; Supplementary Fig. 5h), while in the second half, there was no significant

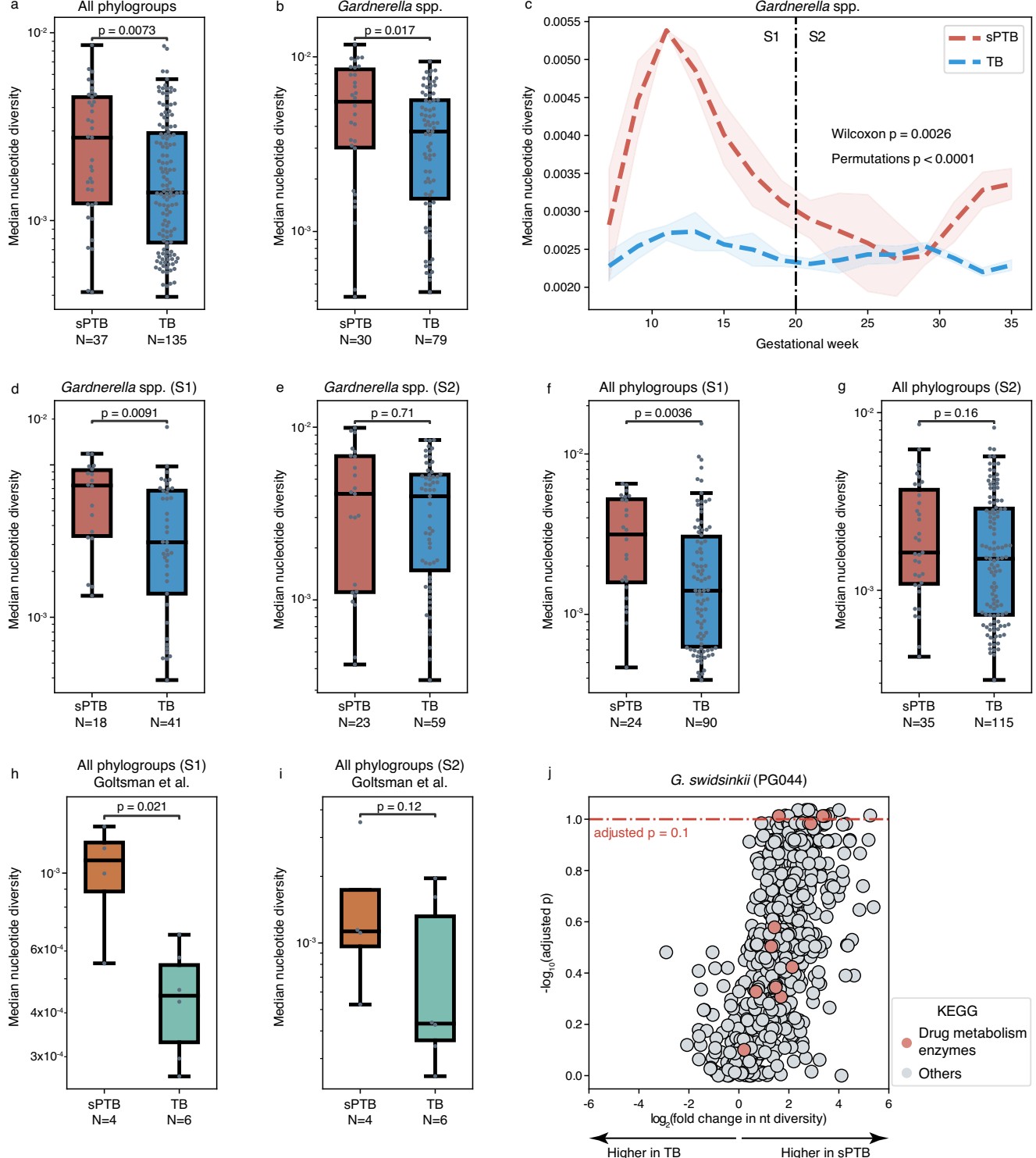

**Fig. 2 | Microdiversity patterns of the vaginal microbiome are associated with sPTB. a**, **b** A comparison of median genome-wide nucleotide diversity along pregnancy between sPTB and TB, displayed for all phylogroups (**a**) and *Gardnerella* spp. (**b**). *p*, two-sided Mann-Whitney. **c** Trajectory of median nucleotide diversity of *Gardnerella* spp. along pregnancy. S1 - first half of pregnancy; S2 - second half of pregnancy. The shaded area depicts mean ± s.d./n. *p*, one-sided Wilcoxon and permutation. **d**, **e** A comparison of median genome-wide nucleotide diversity of *Gardnerella* spp. between sPTB and TB, displayed for pregnancy S1 (**d**) and S2 (**e**). *p*, two-sided Mann-Whitney. **f**, **g** A comparison of median genome-wide nucleotide

diversity of all phylogroups between sPTB and TB, displayed for pregnancy S1 (**f**) and S2 (**g**). *p*, two-sided Mann-Whitney. **h**, **i** Same as **d**–**e**, using data from the independent cohort of Goltsman et al.[32]. *p*, one-sided Mann-Whitney. **j** Volcano plot illustrating the significance (two-sided Mann-Whitney; y-axis) of difference between nucleotide diversity (fold change; x-axis) in sPTB and TB of every gene in *G. swidsinkii* (PG044). Genes above the red dashed line have *P* < 0.05 and an adjusted *P* < 0.1. Genes belonging to KEGG pathways that were significantly enriched in genes showing significant nucleotide diversity differences are color-coded (adjusted *P* < 0.1). Box, IQR; line, median; whiskers, 1.5*IQR.

difference ($p = 0.22$; Supplementary Fig. 5i). We further subsampled an identical number of reads (5000) mapped to *Gardnerella* spp. from each sample and repeated the analysis in Fig. 2b. A significant higher nucleotide diversity in sPTB pregnancies in *Gardnerella* spp. is still detected ($p = 0.028$; Supplementary Fig. 5j), indicating this association is not biased by higher coverage of reads mapped to *Gardnerella* spp. Overall, these results confirm that the sPTB-associated nucleotide diversity we observed was not biased by technical artifacts.

### Evolutionary forces acting on *Gardnerella* species are associated with pregnancy outcomes

Adaptation should increase the fitness of an organism, its ability to survive and reproduce in a given environment. To better understand if the *Gardnerella* spp. populations with higher genetic diversity grow better in the vaginal environment associated with sPTB, we inferred fitness using two measures: relative abundance and growth rate. Indeed, we found that nucleotide diversity in these species was positively correlated with relative abundance (Spearman $\rho = 0.35$, $p = 0.0013$; Fig. 3a). This correlation was not observed in other phylogroups (Supplementary Fig. 6a). *L. crispatus* (PG080) and *L. iners* (PG086) even showed a significantly negative correlation ($\rho = -0.39$ and $-0.32$, $p = 0.026$ and 0.0014, respectively; Supplementary Fig. 6b, c). We additionally used gRodon[36] to predict the maximal growth rate of microbes based on codon usage bias in highly expressed genes encoding ribosomal proteins. We found that in the first half of pregnancy, *Gardnerella* spp. had a somewhat higher, albeit not statistically significant, maximal growth rate in sPTB pregnancies (Mann-Whitney $p = 0.057$; Fig. 3b), while in the second half of pregnancy, the difference was diminished ($p = 0.15$; Fig. 3c). No significant difference in the maximal growth rate was observed in non-*Gardnerella* phylogroups ($P > 0.05$ for all). The in situ growth rates[37,38] of *Gardnerella* were not significantly different between the groups ($p = 0.14$). These results suggest that the sPTB-associated genetic diversity observed in *Gardnerella* spp. may be related to the optimization for faster growth in the sPTB-associated vaginal environment.

Microbial population structure is influenced by various evolutionary processes including selection and homologous recombination[39]. Competence, a mechanism of horizontal gene transfer which involves homologous recombination, has been identified in *Gardnerella* spp[40]. To better interpret the significant differences we observed in the microdiversity patterns of *Gardnerella* spp. between sPTB and TB, we quantified the degree of homologous recombination using the normalized coefficient of linkage disequilibrium between alleles at two loci, D'. A value of D' closer to 0 indicates a higher degree of recombination[41]. Interestingly, we found that the median D' of *Gardnerella* spp. was significantly smaller in sPTB pregnancies in both the first (Mann-Whitney $p = 0.041$; Fig. 3d) and second halves of pregnancy ($p = 0.013$; Fig. 3e), and the same was also observed for the D' of three specific *Gardnerella* spp., *G. piotti* (PG042), *G.* gs7 (PG046), and PG047, in the first half of pregnancy ($P < 0.05$, adjusted $P < 0.1$ for all; Supplementary Fig. 7a). No significant difference in recombination was observed in non-*Gardnerella* phylogroups (adjusted $P > 0.1$ for all). These results suggest that *Gardnerella* spp. tends to have more frequent recombination in women who delivered preterm during both halves of pregnancy.

Next, we quantified the degree of selection using dN/dS in this species (Methods). This measure quantifies the ratio between synonymous and non-synonymous mutations, and hence offers insight into the type of selection, with values close to zero indicating purifying selection, and values higher than one indicating positive selection[42]. dN/dS is calculated in relation to the reference, and can therefore detect selection on mutations that have already been fixed within the population[43]. Consistent with the gut and ocean microbiomes[44–46], purifying selection is predominant across all genes of the vaginal microbiome (dN/dS $\ll$ 1; median [IQR] dN/dS of 0.17 [0.10, 0.29];

Supplementary Fig. 7b). While the median dN/dS of all *Gardnerella* spp. genes was not significantly different between sPTB and TB pregnancies (Mann-Whitney $U$ $p = 0.48$), we detected some differences when examining high-level functions (COG categories[47]) within each half of pregnancy. In the first half, the median dN/dS of *Gardnerella* spp. genes was somewhat lower in sPTB pregnancies for inorganic ion transport and metabolism, lipid transport and metabolism, secondary structure, and cell wall/membrane/envelope biogenesis, though this was not statistically significant after adjusting for multiple testing (COG categories "P", "I", "Q", and "M", respectively; Mann-Whitney $P < 0.05$, adjusted $P > 0.1$ for all; Supplementary Fig. 7c). In the second half of pregnancy, the median dN/dS was significantly lower in sPTB pregnancies for lipid transport and metabolism and cell motility (COG categories "I" "N"; Mann-Whitney $p = 0.0040$ and $p = 0.04$, adjusted $p = 0.07$ and 0.40, respectively; Fig. 3f). No significant difference in the selection based on dN/dS was observed in non-*Gardnerella* phylogroups ($P > 0.05$ for all). Our results suggest that *Gardnerella* spp. genes involved in lipid transport and metabolism may undergo stronger purifying selection in sPTB. As purifying selection maintains the fitness of organisms by constantly sweeping away deleterious mutations and conserving functions, *Gardnerella* spp. may benefit from this stronger purifying selection targeting lipid functioning when growing in the sPTB-associated vaginal environment during pregnancy.

### sPTB-associated vaginal microbiomes have a higher antibiotic-resistance potential

Antibiotics are widely used during pregnancy, sometimes even topically in the vagina[48]. This exposure may promote antimicrobial resistance (AMR). To assess if antibiotic-resistance potential in the vaginal microbiome is associated with sPTB, we subsampled an identical number of reads ($10^5$) from each sample and mapped them to the Comprehensive Antibiotic Resistance Database[49]. The total number of reads mapped to AMR reference genes was significantly higher in the first half of sPTB pregnancies (Mann-Whitney $U$ $p = 0.015$; Fig. 4a), but not in the second half ($p = 0.76$; Fig. 4b). In addition, to assess the difference of specific AMR genes between the vaginal microbiomes of sPTB and TB, we identified AMR genes in the genomic assemblies. A significantly higher median count and Shannon-Wiener diversity of AMR genes were detected in vaginal microbes sampled at the first half of pregnancies that ended preterm (3-times higher on average; Mann-Whitney $U$ $p = 0.011$ and $p = 0.0078$, respectively; Fig. 4c, e, respectively), yet this difference was not detected in the second half ($p = 0.16$ for both; Fig. 4d, f, respectively). Exploring the source of these genes, we found a significantly higher median fraction of phage-borne AMR genes in the microbiomes of women who delivered preterm (one-sided $p = 0.016$, adjusted $p = 0.093$.; Fig. 4g), suggesting transduction may promote the higher median count and diversity of AMR genes observed in the first half of sPTB pregnancies (Fig. 4c, e). Among the 9 AMR gene categories that had genes present in at least 10% of women, phenicol, aminoglycoside, glycopeptide, and MLS resistance genes showed a significantly higher median fraction in the sPTB microbiome ($P < 0.05$, adjusted $P < 0.1$ for all; Fig. 4h). These results suggest a unique antibiotic resistance profile associated with the first half of sPTB pregnancies, potentially indicative of usage of specific antibiotics. Indeed, we detected a somewhat higher richness of AMR genes along the first half of pregnancy in the 29 women who used antibiotics in the past 6 months before pregnancy than those who did not (Mann-Whitney $U$ $p = 0.079$; Fig. 4i). This is also consistent with our observation that genes with sPTB-associated nucleotide diversity were enriched for drug metabolism in *G. swidsinskii* (PG044) (Fig. 2d).

As a higher risk of preterm birth has been reported to be associated with bacterial vaginosis (BV)[50–52], and antibiotics used to treat BV may change the AMR genetic profiles of the vaginal microbiome, our findings in AMR genes might be potentially confounded by BV. To

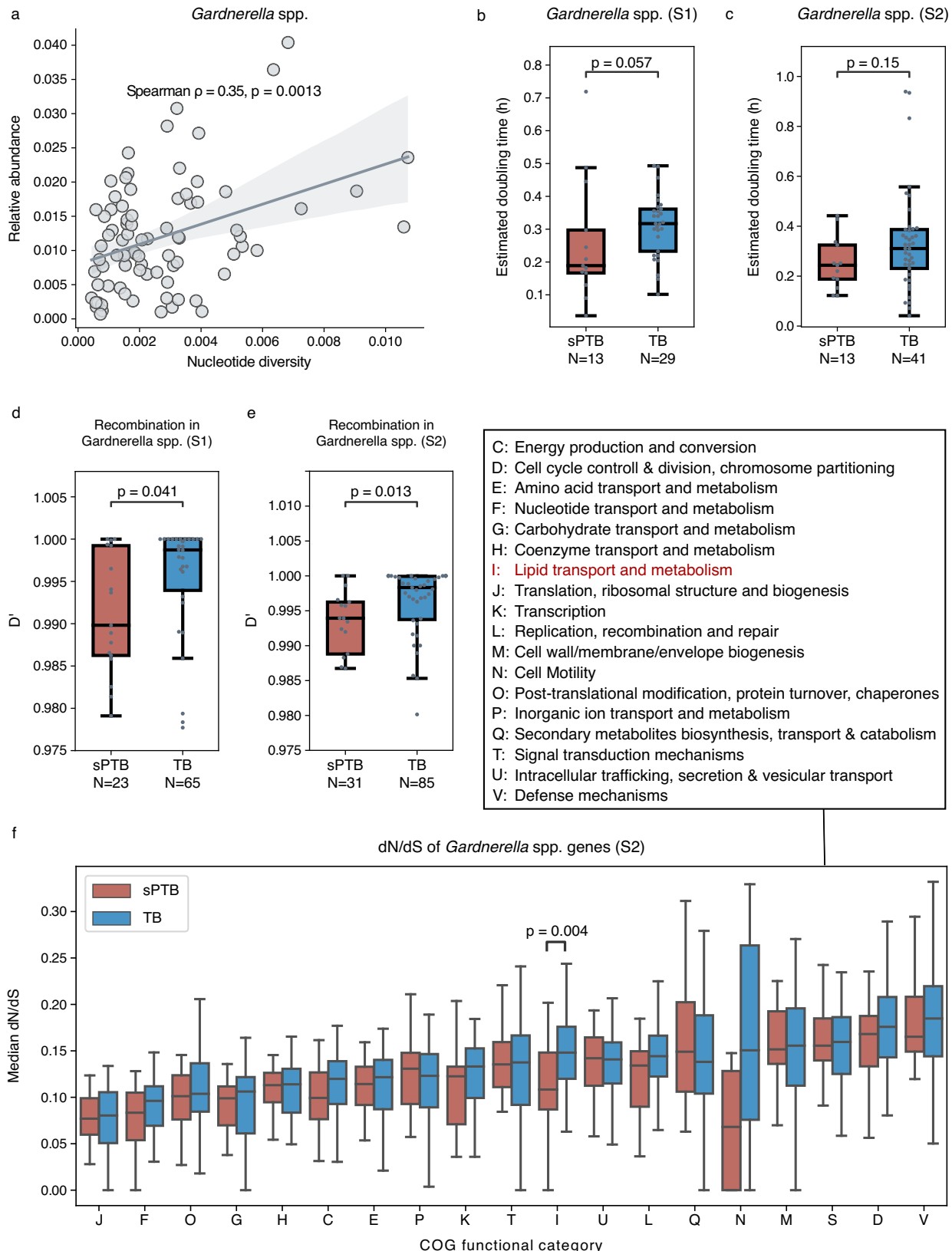

**Nature Communications** | (2023)14:4997

assess this potential bias, we compared AMR gene count between women with and without BV as well as between women with and without BV history. We found that the AMR gene count was not significantly different between women with and without BV for both halves of pregnancy ($p = 0.38$ and 0.5, respectively; Supplementary Fig. 8a). Similarly, it was not different between women with and

without BV history ($p = 0.45$ and 0.1, respectively; Supplementary Fig. 8b). These results suggest that the association between AMR gene and pregnancy outcomes is independent of BV.

To check if the strong association between sPTB and the AMR potential of the vaginal microbiome is contributed by a particular phylogroup, we performed a similar analysis for each phylogroup. We

**Fig. 3 | Evolutionary forces on *Gardnerella* spp. a** Spearman correlation between median genome-wide nucleotide diversity and relative abundance of *Gardnerella* spp. along pregnancy. The line and the shaded area depict the best-fit trendline and the 95% confidence interval (mean ± 1.96 s.e.m.) of the linear regression. **b, c** Predicted maximal doubling time (gRodon[36]) of *Gardnerella* spp. compared between sPTB and TB, displayed for the first (**b**, S1) and second (**c**, S2) halves of pregnancy. **d, e** Median D′ of *Gardnerella* spp compared between sPTB and TB, displayed for the first (S1, **d**) and second (S2, **e**) halves of pregnancy. Lower D′ indicates more frequent recombination. **f** dN/dS of *Gardnerella* spp genes compared between sPTB and TB by COG functional categories, displayed for the second half of pregnancy (S2). dN/dS closer to 0 indicates stronger purifying selection. $N = 40$ and 135 for sPTB and TB, respectively, included in this panel. Box, IQR; line, median; whiskers, 1.5*IQR; *p*, two-sided Mann-Whitney.

found, however, that none of them showed a significant difference in the median count and diversity of AMR genes between sPTB and TB (Mann-Whitney *U P* > 0.05 for all). This result suggests that the higher AMR potential associated with sPTB may be a property of the vaginal microbiome as an ecosystem. However, this lack of association could also be driven by underestimation of AMR genes due to the limitation of MAG binning methods in recovering mobile genetic elements[53].

## Discussion

Microbial genomes can exhibit large variations even within the same species, as a result of adaptation to various environments[54]. Associations between the vaginal microbiome and preterm birth have been widely reported[7,8,12,35,52]. However, there is still much left to explore regarding potential mechanisms underlying host-microbiome interactions in this context. Here, by leveraging publicly available metagenomic data[6], we provide a population genetic view of the vaginal microbiome during pregnancy. We identify a number of microbial features including population nucleotide diversity, selection metrics, and antibiotic resistance potential that are associated with sPTB. Interestingly, we find that the higher population nucleotide diversity is driven by *Gardnerella* spp. during the first half of pregnancy. This species appears to undergo more intense changes in the population structure contributed by recombination and purifying selection in pregnancies which ended preterm. We also show evidence that this sPTB-associated genetic pattern of *Gardnerella* spp. may be related to optimization of growth rates in vaginal conditions linked to sPTB. Our results are indicative of adaptation of the vaginal microbiota to the host, which in turn may influence pregnancy outcomes.

Our findings regarding a relationship between ecological processes in the pregnancy vaginal microbiome and subsequent preterm birth are consistent with previous studies[6,7,11,12,14,52,55]. We add to these studies by exploring an additional layer of microbial variability associated with sPTB - microbial genetic diversity. It is known that genomic variation within species can result in phenotypic diversity and adaptations to different environments[54]. These adaptations, in turn, can affect host phenotypes such as disease outcomes[56]. Such associations between microbial genomic variation and host phenotypes have been reported in the gut microbiome[45,57,58]. Our study suggests that this phenomenon also occurs in the vaginal ecosystem, and that it may be associated with pregnancy outcomes. Nethertheless, the associations between microbial genetic diversity and pregnancy outcomes we detect might also be a consequence of different processes or unmeasured confounders that act on both variables (e.g., certain drugs or exogenous chemical compounds are that drive inflammation), and while we find this unlikely, this should be determined by future studies.

Interestingly, we found that the association of genetic diversity and sPTB was largely driven by *Gardnerella* spp., a group of species commonly associated with BV[50–52]. A number of studies reported a higher abundance of these species in sPTB pregnancies[6,9,52,59,60]. A recent preprint also demonstrated a higher number of *Gardnerella* clades in sPTB[61]. We show that *Gardnerella* spp. populations with more genetically diverse strains may also be associated with sPTB. In addition, we found that this taxon has the capacity to grow 1.5 times faster in pregnancies that ended preterm, consistent with an overall higher relative transcriptional rate of *G. vaginalis* which was previously reported[6]. These more genetically diverse strains appear to have adapted to the vaginal environment associated with sPTB, exhibiting

higher fitness. Notably, the higher genetic diversity associated with sPTB in *Gardnerella* spp. was detected during the first half of the pregnancy (<20 gestational week) rather than the second half. The enriched nucleotide diversity during the first half of pregnancy might be related to the change of human chorionic gonadotropin (HCG), which peaks at roughly the same time as *G. vaginalis* microdiversity as we observed in Fig. 2c, and was suggested to play an immunomodulatory role in humans[62,63]. To our knowledge, however, the effect of HCG on the vaginal ecosystem is not well established yet. Most potential biomarkers of sPTB (e.g., serum alpha-fetoprotein[64]) were so far identified using samples from the second trimester of pregnancy (gestational week 14–27). Our results suggest that high resolution analysis of microbiome samples from even earlier stages of pregnancy (<week 20) may yield informative biomarkers of pregnancy outcomes.

As in the human gut microbiome[19–21], we show evidence that adaptive evolution also occurs in the vaginal microbiome. Several environmental factors affecting the vaginal ecosystem, such as pH, neutrophil levels, and xenobiotics, have been reported to be associated with sPTB[35,65]. These environmental factors may act as selective stressors that lead to different evolutionary patterns in the vaginal microbiome. Indeed, we detected more frequent homologous recombination and stronger purifying selection within *Gardnerella* spp. during pregnancies that end preterm. Homologous recombination is a critical mechanism speeding adaptation by increasing fixation probability of beneficial mutations[66] and reducing clonal interference (i.e., competition between beneficial mutations) in bacteria[67]. Purifying selection also contributes to adaptation by sweeping away deleterious mutations and conserving functions, such as in oligotrophic nutrient conditions[44,68]. Notably, we found that sPTB-associated purifying selection is particularly strong on genes involved in lipid transportation and metabolism. This is consistent with previous identification of lipid metabolites (e.g., monoacylglycerols and sphingolipids) as signatures of sPTB[35,69]. Whether this stronger purifying selection targeting lipid transportation and metabolisms in pregnancy that ended preterm leads to changes in the concentrations of lipid metabolites however requires further experimental testing. As both recombination and purifying selection can reduce genetic diversity, sPTB-associated recombination and purifying selection along pregnancy may explain the higher nucleotide diversity of *Gardnerella* spp. in sPTB in the first half of pregnancy compared to the second half.

Antibiotics are common selective stresses acting on the human microbiome[29] and have been associated with preterm birth[70]. We detected higher count and diversity of AMR genes associated with sPTB, which our analysis suggests to be facilitated by prophages in preterm vaginal microbiomes. While multiple phages (e.g., Siphoviridae, Myoviridae, and Microviridae) have been detected in the vagina of pregnant women, their association with sPTB is rarely studied[71]. Our results imply a potentially important role of bacteria-phage interactions in pregnancy outcomes via transferring of AMR genes. We also found that genes related to phenicol and aminoglycoside resistance were more abundant in vaginal microbiomes during pregnancies that ended preterm. While both antibiotics have been frequently used to treat gynecologic infection for decades[72], and some phenicols (e.g., chloramphenicol) are thought to be safe for use during pregnancy[73] aminoglycoside is teratogenic. Previous studies reported that exposure to antibiotics could change the composition of the vaginal microbiome[48,74], indicating an ecological effect. In comparison, our

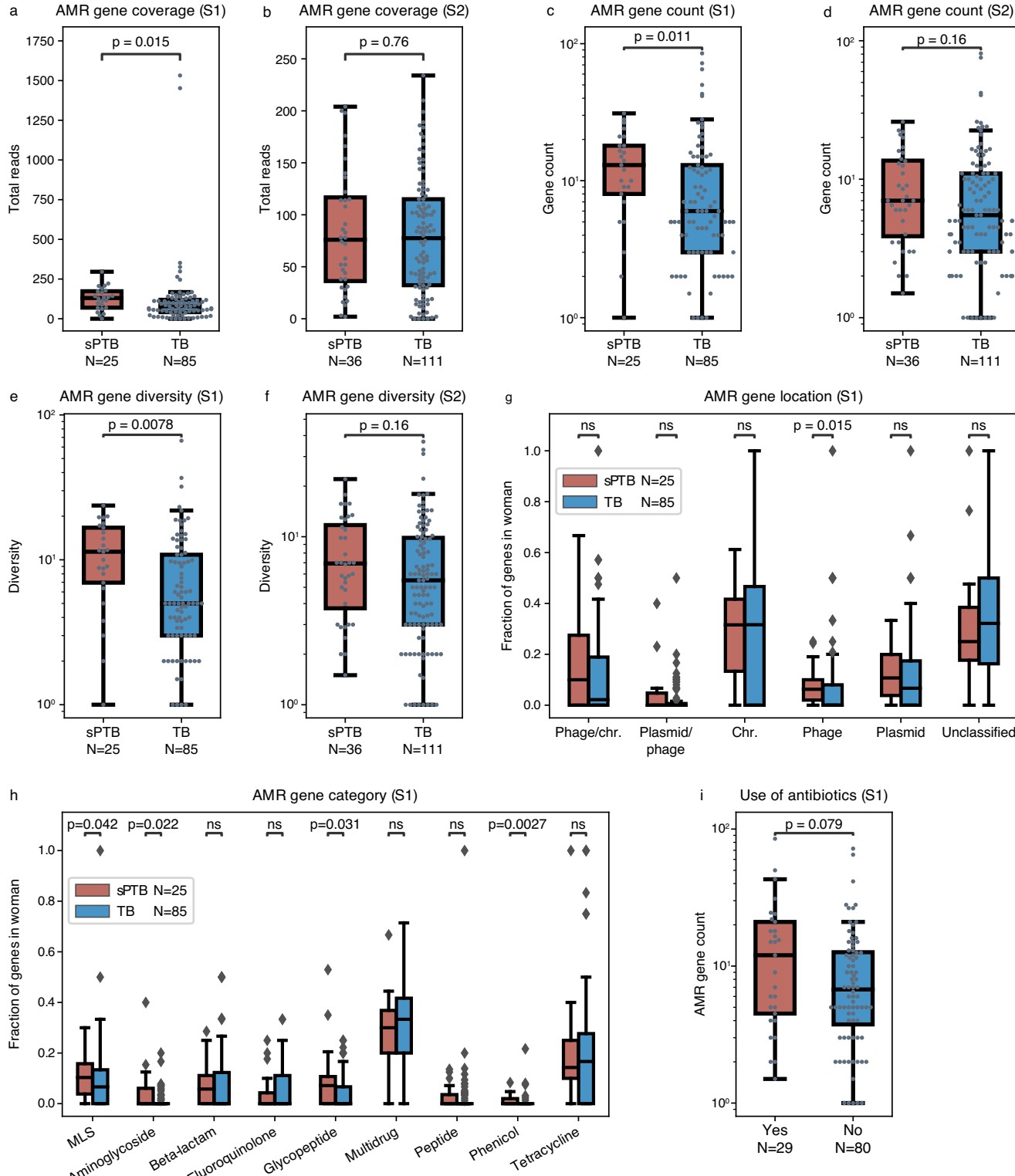

**Fig. 4 | Antimicrobial resistance (AMR) gene profiles of the vaginal microbiome are associated with sPTB. a, b** Total subsampled reads ($10^5$) mapped to AMR genes compared between sPTB and TB, in the first (S1, **a**) and second (S2, **b**) halves of pregnancy. **c, d** Median count (along period) of AMR genes compared between sPTB and TB, in the first (S1, **c**) and second (S2, **d**) halves of pregnancy. **e, f** Median Shannon-Wiener diversity (along period) of AMR genes compared between sPTB and TB, in the first (S1, **e**) and second (S2, **f**) halves of pregnancy. **g** Fraction of AMR genes originating in different locations, shown as median along the first half of each pregnancy. Chr.: chromosome. **h** Fraction of AMR genes belonging to different resistance categories, shown as median along the first half of each pregnancy. MLS, macrolide, lincosamide and streptogramin B. **i** AMR gene richness in the first half of pregnancy (S1) compared between women who used and did not use antibiotics in the 6 months before pregnancy. Box, IQR; line, median; whiskers, 1.5*IQR; *p*, two-sided Mann-Whitney *U*.

results may suggest adaptation of the vaginal microbiome to more frequent antibiotics usage in women who delivered preterm, leading to an enrichment of AMR genes as well as higher nucleotide diversity in *Gardnerella* spp. genes encoding enzymes for drug metabolism. While this hypothesis requires further study, it is further supported by the fact that a higher proportion of women who delivered preterm (31%) had used antibiotics in the past 6 months before pregnancy than women who delivered at term (23%).

Despite its findings, our study is limited by low sequencing depth (median bacterial read count $<5 \times 10^5$) and inconsistent sampling frequency during pregnancy (1 to 8 samples per pregnancy, with an average of 4). These limitations lead to high sparsity in the features analyzed, preventing a more in-depth temporal and predictive analysis of the link between population genetics of the vaginal microbiome and sPTB. Our results warrant a high-resolution investigation of the vaginal metagenome, with frequent sampling and high sequencing depth.

In summary, through in-depth population genomic analyses, our study identified genetic and functional associations between the vaginal microbiome and preterm birth. We revealed evidence of microbial genetic adaptation to the host environment linked to preterm birth and highlighted the importance of microbial evolutionary processes to adverse pregnancy outcomes, particularly in *Gardnerella* spp. Future investigation on the pressures driving the sPTB-associated microbial adaptation is warranted to fully understand the molecular mechanisms underlying preterm birth.

## Methods

### Sample selection and metagenomic data

We analyzed published metagenomic sequencing data from the Multi-Omic Microbiome Study: Pregnancy Initiative (MOMS-PI) PTB case-control study[6] as well as publicly available data from ref. 32 for validation. The metagenomic sequencing data and clinical covariates from ref. 6 were obtained from dbGaP (study no. 20280; accession ID phs001523.v1.p1) and directly from Virginia Commonwealth University. MOMS-PI participants provided written informed consent allowing for their data to be shared and used in future research studies. The analyses performed here were approved by the IRB of Columbia University, approval number AAAS5367.

MOMS-PI recruited a majority of women identifying as Black as described in ref. 6. We obtained 135 vaginal samples collected longitudinally during pregnancy from 40 women who eventually delivered preterm spontaneously (sPTB; mean ± sd) age 26 ± 5.68) and 570 vaginal samples from 135 women who delivered at term (TB; age 25.9 ± 5.43). Samples were sequenced paired-end, to a mean ± sd depth of 717,887 ± 1,536,354 (mean ± s.d.) non-human reads. Fettweis et al. 2019[6] only included in this dataset term births after 39 weeks of gestation, with the intention of avoiding complications associated with early term birth[6]. Our study therefore uses the same definitions: spontaneous preterm birth is defined as live birth between 23 and 37 gestational weeks without medical indication, and term birth is defined as live birth at or after 39 gestational weeks.

On average, 3.36 and 3.21 samples were collected for each sPTB and TB woman, respectively (Mann-Whitney $U$ $p = 0.83$); 1.68 and 1.51 samples were collected for the first half of pregnancy for each sPTB and TB woman, respectively ($p = 0.30$); and 2.34 and 2.14 samples were collected for the second half of pregnancy for each sPTB and TB woman, respectively ($p = 0.27$) (Supplementary Table 1). The average gestational age at the first sample being collected for sPTB and TB women is 17.38 and 16.09, respectively ($p = 0.45$), while for the last sample being collected for sPTB and TB women is 31.23 and 32.31, respectively ($p = 0.72$) (Supplementary Table 1). In addition, none of these women delivered at <20 gestational weeks (Supplementary Fig. 9a). These indicate that dropping out due to late miscarriage/early PTB is not a concern to bias our findings in this study.

To check for the presence of some potential confounders for vaginal microbiome-sPTB associations, we calculated propensity scores[75] for each subject based on income, age, and race using a logistic regression model. We found that propensity scores for both sPTB and TB subjects exhibited a similar distribution (Kolmogorov–Smirnov test $p = 0.21$), suggesting the associations we detect are not likely to be confounded with these variables (Supplementary Fig. 9b). These results suggest a negligible confounding effect of income, age, and race in this study on microbiome-sPTB associations. We note, however, that population studies such as the one performed by Fettweis et al.[6] can be exposed to selection bias, via access to medical care and other reasons. Experimental procedures for data generation are described by Fettweis et al[6].

### Metagenomic assembly, genomic binning, genome annotation, and relative abundance

Our analysis follows the accepted standards used in refs. 76–79, using the ATLAS pipeline[80] (v. 2.4.4). Bases with quality scores <25, raw reads <50 bp lengths, and sequencing adapters were removed using Trimmomatic v.0.39[81]. Reads mapped to human and PhiX genome sequences were removed by mapping with Bowtie2 v.2.3.5.1[82]. Assembly and binning were done with ATLAS: filtered reads were assembled using metaSPAdes v.3.15.2[83], and contigs were binned into metagenome-assembled genomes (MAGs) using MetaBAT2 v.2.14.0 (ref. 84) with a minimum contig length of 1500. Quality, GC content, genome size, and taxonomy of MAGs were estimated using CheckM v.1.0.9[24]. MAGs were de-replicated using dRep v.3.2.0[85] with an average nucleotide identity (ANI) of 0.95, minimum completeness of 50%, and maximum genome contamination of 10%. The MAG with the highest dRep score within each 95% ANI cluster, termed here as a phylogroup, was selected as the representative MAGs for that phylogroup. Genes were predicted using Prodigal v.2.6.3[86] and annotated using EggNOG v.5.0[87]. Filtered reads were mapped to representative MAGs using Bowtie2 v.2.3.5.1[88]. The relative abundance of each representative MAG was calculated by dividing the number of reads that mapped to that MAG, corrected to the genome size and completeness, by the total number of reads in each sample.

### Phylogeny, ANI, and dendrogram

Amino acid (AA) sequences of 120 marker genes were called and aligned for representative MAGs using GTDB-Tk v.1.5.1 (ref. 89). MAGs with <60% of AA in the alignment were excluded in the phylogenetic tree construction. The best evolutionary model LG + G + I (the Le Gascuel model + gamma distribution + invariant sites) was identified using prottest3 v.3.4.2 (ref. 90) and 500 bootstraps were used for tree construction using RAxML v.8.2.12 (ref. 91). The tree was rooted by midpoint and visualized in iTol v.6.3 (ref. 92).

Pairwise ANI of MAGs for each phylogroup as well as for representative MAGs annotated as *G. vaginalis* (PG42-53) and reference genomes of 13 *Gardnerella* species defined in ref. 25 was calculated using pyani v.0.2. Dendrogram of ANI was constructed using the complete-linkage clustering method in the vegan package (v.2.6-4) in R v.3.6.0.

### Microdiversity profiling, growth rate estimation, and antimicrobial resistance genes

Population microdiversity metrics including genome-wide nucleotide diversity, gene-wide nucleotide diversity, linkage disequilibrium measures (D') and selection measures (dN/dS), were calculated using InStrain v1.0.0[43] using the 157 representative MAGs as the reference database. Maximal growth rate was estimated for each MAG using gRodon[36] v.1. Antimicrobial resistance (AMR) genes were detected in assemblies and MAGs using PathoFact v.1.0 (ref. 93) with default parameters.

## Temporal analysis

To generate the trajectories representing the change in nucleotide diversity over time in term and preterm deliveries, we pooled the temporal data of *Gardnerella* spp. from all women in each group (term and preterm). When we had more than one observation per gestational week, we took the median value across samples. We then binned the temporal data into bins of 3 weeks, except for the first bin which spanned weeks 1-7, and took the median of each bin as a summary. To smooth the observed binned data we applied splines, which is a special function defined piecewise by polynomials for data smoothing. To compare between the temporal trajectories of preterm and term, we performed a permutation test, in which we generated a null distribution of euclidean distances by shuffling the these trajectories $10^4$ times and comparing to the euclidean distance in the original data[31].

## Functional enrichment analysis

To identify COG/KEGG pathways that were enriched in genes showing significant difference in nucleotide diversity between sPTB and TB, the frequency of each COG/KEGG category was first calculated from significant genes (observed frequency). Then, the frequency of each COG/KEGG category was calculated from an identical number of genes randomly selected from all genes (expected frequency). This process was repeated 10,000 times. The null hypothesis was that the observed frequency of COG category is smaller than the expectation. For each COG, probability $P$ of the null hypothesis was calculated using the formula: $p = |[x_i \in \mathbf{x}: x_i > = k]| / 10000$, where $[...]$ denotes a multiset, $\mathbf{x} = (x_1, x_2, ..., x_n)$ is a list of expected values, and $k$ is the observed value.

## Statistics & reproducibility

This was a secondary analysis of data obtained from a case-control analysis of an observational cohort[6]. Sample size follows the original study and no sample size calculation was performed for this study. No data was excluded from this analysis. As this was a secondary analysis of an observational study, there was no allocation to intervention, and hence no randomization or blinding to allocation. A STORMS checklist is provided as Supplementary Data 3. A different number of samples was available for each woman in the database. In our analyses, we therefore used the median along pregnancy (or its first or second half). The false-discovery rate procedure (FDR) of Benjamini and Hochberg (BH)[94] was used to correct for multiple testing. Adjusted $P < 0.1$ was used as the significance cutoff.

## Reporting summary

Further information on research design is available in the Nature Portfolio Reporting Summary linked to this article.

## Data availability

The raw sequencing data and metadata were obtained from ref. 6. Data are available under restricted access for ethical and privacy concerns from dbGAP under accession phs001523. Access can be obtained by a data access request to dbGaP, under the purview of the data access committee of the Eunice Kennedy Shriver National Institute of Child Health and Human Development [HD-DAC@mail.nih.gov]. All processed features generated in this study are available[95] at Zenodo, https://doi.org/10.5281/zenodo.8150902. The validation dataset[32] is available in SRA, under accession PRJNA288562.

## Code availability

Code to replicate all analyses is available[96] from https://github.com/korem-lab/MOMs-PI_microdiversity_2023.

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

## Acknowledgements

We thank members of the Korem lab for useful discussions. This study was supported by the Eunice Kennedy Shriver National Institute of Child Health and Human Development (NICHD) of the National Institutes of Health under award number R01HD106017, the Program for Mathematical Genomics at Columbia University, and the CIFAR Azrieli Global Scholarship in the Humans & the Microbiome Program (T.K.). The dataset used was obtained from dbGaP (phs001523), using data provided by Gregory A. Buck, Ph.D. and colleagues and supported by NICHD (U54 HD080784) (G.A.B.).

## Author contributions

J.L. and T.K. designed the study. J.L., L.S., and J.A.U analyzed the data with input from T.K., M.S., B.Z and G.A.B. J.L. wrote the manuscript with input from L.S., T.K., M.S., B.Z. and G.A.B. G.A.B. assisted with data access and acquisition.

## Competing interests

G.A.B. is a member of the Scientific Advisory Board of Juno, LTD., a startup biotech firm focused on using the vaginal microbiome to address issues of women's gynecologic and reproductive health. Juno had no involvement in the current study. Other authors declare no competing interests.
