## [Peer Review File · Nature Communications]

REVIEWER COMMENTS

Reviewer #1 (Remarks to the Author):

The authors present a very nice analysis of vaginal microbiome during pregnancy - both normal as well as women who spontaneously deliver preterm. The question is important and valuable and the analysis is overall robust. There are a few suggestions to address:

1) It is unclear whether the women had a single or multiple samples taken and when during the pregnancy that took place. Table 1 with all the sample / participant information would be helpful.

2) How is the longitudinal nature of the data taken into account during analysis? If the median is taken, it might not be representative unless the authors show that the microbiome is stable throughout pregnancy.

3) The authors focus on the gardnerella findings, but the result so the overall analysis should also be reported. What about overall diversity? What about the associations for other species even if they are known?

4) The study would be much stronger if independent validation was carried out in another cohort. Is there a cohort from the public domain that could be utilized to confirm the presented findings?

5) All the code to do the analysis should be made available.

Reviewer #2 (Remarks to the Author):

This is secondary analysis of existing metagenomic data sets from the MOMs study (175 women term and 40 sPTB with longitudinal sampling). It focuses on expanding on the previous analysis and micro-diversity. The genetic analysis of the Gardnerella vaginalis species/groups is fascinating and timely (given the burgeoning interest in the literature). AMR gene analysis is highly relevant although given we know that women with BV are at higher risk of PTB it is not surprising that the sPTB group a strong association between of AMR potential detected.

Comments

Please comment on why the sampling groups has a gestation gap with PTB being women delivering up to 37 weeks, but term women only selected from 39 weeks onwards. Also reflect on how this approach (and worries about overlap between late preterm and early term data) impacts on the clinical relevance of the information.

Comment on how findings in this refined analysis potentially modifies conclusions drawn from previous analyses.

A slight issue that I had with the original paper, which may still be pertinent to the current paper is that there is no mention on how clinical management may have affected measurements of the vaginal microbiota, especially longitudinally. How many of the women received interventions for risk of preterm birth e.g. cervical suture, progesterone etc. These may have altered the microbiome composition across gestation (inflammatory responses change with intervention).

Differences detected between the first half of pregnancy versus the second half of the pregnancy are reported – please can you confirm that they had samples from all the women in both time periods? Otherwise drop out (potentially linked to late miscarriage/early PTB could bias the data. Also should suggest why there might be a difference in the discussion - is it that the hormonal milieu of pregnancy in the second half of pregnancy is beneficial (link in with other observations that the microbiome becomes more lactobacillus dominated in later pregnancy?)

Whilst in depth review of analytical approaches is outside of my expertise, they appear to be robust, appropriate, and well explained. It would be useful to know whether the code for any of the analysis is deposited for public access?

Propensity scores were used to assess a limited number of confounders, but as there are other confounders, particularly those that drive inflammation, can affect microbiome composition. It would be useful to discuss these and highlight in the limitations.

The authors in the methods state they account for false discovery rates. Is this undertaken for all the different types of analysis shown (e.g. Fig 4).

Figure legends: It would be useful to give gestational time points studies and n numbers in the figure legends (e.g. Fig 1). I am aware some information provided in methods, but figure legends should give all information needed to interpret the data.

Reviewer #3 (Remarks to the Author):

In this manuscript Lioa et. al. perform detailed metagenomic analyses on a set of previously sequenced stool samples from a cohort of infants born term and preterm. The main goal of the manuscript is to identify microbial signatures associated with preterm birth. The most substantial finding of the work is their identification of Gardnerella microdiversity being associated with preterm birth. Overall I found this manuscript to be technically sound, with some minor comments listed below, but a little difficult to interpret the biological underpinnings of the results. To me this study feels like a great exploratory look into the associations between microdiversity and preterm birth, and it provides solid justification for further research in this area.

Specific comments:

1) A really great deal of work is put into showing that the association between Gardnerella and microdiversity is not due to sampling and sequencing bias, which I applaud the authors for, but I still have concerns that sequencing coverage may be impacting the result. On lines 261 to 271, the authors subset each sample to an identical number of reads, but if Gardnerella is more abundant in preterm birth, this will still result in higher coverage of Gardnerella in PTB samples. If coverage is associated with nucleotide diversity, this would explain the higher nucleotide diversity in PTB samples. A test that would evaluate this directly would be to subset a variable amount of reads from each sample to make Gardnerella have the same sequencing coverage / depth in all samples. This would directly control for any relationship between sequencing coverage and nucleotide diversity.

2) I like the analysis performed in lines 144 - 152, but it's unclear why *M. genomosp.* was the only phylogroup tested. Were all phylogroups tested and this was the only significant one? If so, was the proper FDR correction performed?

3) In the analysis associated with Figure 3b, why did the authors choose to evaluate replication rates using codon bias instead of tools that profile in situ replication rates like iRep and PTR?

- Dr. Matthew Olm

Reviewer #1 (Remarks to the Author):

The authors present a very nice analysis of vaginal microbiome during pregnancy - both normal as well as women who spontaneously deliver preterm. The question is important and valuable and the analysis is overall robust.

We thank the reviewer for their assessment as well as for their recognition of the value and importance of our study question and the overall robustness of our analyses.

There are a few suggestions to address:

1) It is unclear whether the women had a single or multiple samples taken and when during the pregnancy that took place. Table 1 with all the sample / participant information would be helpful.

We thank the reviewer for highlighting this gap. We have added a table (**Supplementary Table 2**; provided below for convenience) summarizing the number of longitudinal samples collected for women who deliver spontaneous preterm (sPTB) and at term (TB) during pregnancy. On average, 3.36 and 3.21 samples were collected for each sPTB and TB woman, respectively; 1.68 and 1.51 samples were collected for the first half of pregnancy for each sPTB and TB woman, respectively; and 2.34 and 2.14 samples were collected for the second half of pregnancy for each sPTB and TB woman, respectively. The average gestational age at the first sample being collected for sPTB and TB women is 17.38 and 16.09, respectively, while for the last sample being collected for sPTB and TB women is 31.23 and 32.31, respectively. The description for this new table has been added to the results and methods sections of the revised manuscript (L98-99, 522-528).

Supplementary Table 2. Summary of longitudinal samples collected during pregnancy for women who deliver spontaneous preterm (sPTB) and at term (TB).

	sPTB	TB	Mann-Whitney U P
N of samples / woman (mean \pm s.d.)	3.36 \pm 1.51	3.21 \pm 1.13	0.83
N of samples / woman for the first half of pregnancy (mean \pm s.d.)	1.68 \pm 0.69	1.51 \pm 0.56	0.30
N of samples / woman for the second half of pregnancy (mean \pm s.d.)	2.34 \pm 1.10	2.14 \pm 0.90	0.27
Gestational age at the first sample (mean \pm s.d.)	17.38 \pm 8.08	16.09 \pm 7.50	0.45
Gestational age at the last sample (mean \pm s.d.)	31.23 \pm 5.52	32.31 \pm 3.66	0.72

2) How is the longitudinal nature of the data taken into account during analysis? If the median is taken, it might not be representative unless the authors show that the microbiome is stable throughout pregnancy.

We thank the reviewer for this comment. While the metagenomic data that we obtained from the iHMP MOMS-PI cohort offers an opportunity for a deep investigation of the vaginal microbiome, it also has its limitations. Specifically, and likely due to the high fraction of human reads in vaginal swabs, many samples have low sequencing depth (~75% of samples have less than 10^6 reads; **see Reviewer Fig. 1**). This causes sparsity for some of the metrics analyzed, which does not allow for robust temporal analysis using each gestational week

as a time point, and necessitates measures such as using the median. Nevertheless, while we agree that a median might not fully capture microdiversity at every specific point in time, we claim that our separate considerations of the two halves of pregnancy does capture some temporal aspect of the microbiome along pregnancy. In any case, the measures we calculate do offer an informative representation of properties of the vaginal microbiome. This is demonstrated by the robust associations that we found, which we now also replicate in an independent cohort (**Fig. 2h,i**). We discuss limitations with respect to sequencing depth and sparsity in the discussion (L493-498).

Additionally, we would like to point out that we have performed several temporal analyses using longitudinal data during pregnancy in this study. First, we analyzed temporal changes in the microdiversity of *G. vaginalis* along pregnancy for the sPTB and TB groups (**Fig. 2c**). We found that while the microdiversity of *G. vaginalis* in the TB group is relatively stable during pregnancy, it increased significantly in the first half of pregnancies in the sPTB group. Both groups had stable microdiversity in the second half of pregnancy. As the significant difference between sPTB and TB is observed in the first half of pregnancy, we separated subsequent analyses to the two halves of pregnancy, which we believe captures a significant aspect of the temporal signal. Second, we used compositional tensor factorization (CTF) to investigate if the temporal dynamics of the composition of the vaginal microbiome is associated with sPTB, an analysis based on longitudinal data (**Fig. 1b, c, d**). Finally, in the revised manuscript, we added an additional analysis demonstrating that the vaginal microbial signatures of term and preterm pregnancies are separated throughout pregnancy, with more prominent changes during the first two trimesters (**Fig. 1e**; provided below for convenience).

Reviewer **Fig. 1** Distribution of read count of samples.

Fig. 1 | The composition of the vaginal microbiome associated with sPTB. e. Log ratio of top taxa from (d) over time, separated to PTB and TB samples. Shaded area, 95% CI.

3) The authors focus on the *Gardnerella* findings, but the result so the overall analysis should also be reported. What about overall diversity? What about the associations for other species even if they are known?

We thank the reviewer for this comment. While we have focused some of our analyses on *Gardnerella* (namely, **Fig. 3**), the majority of our analyses were applied systematically for all phylogroups (species):

- We started from an analysis of temporal dynamics of the vaginal microbiome during pregnancy, which was done for all microbes (**Fig. 1**).
- We then proceeded to compare the intraspecies diversity of all species (**Fig. S2**), observing a significant difference in *M. genomosp.*
- Next, we indeed performed an analysis of the overall microdiversity, which the reviewer directly asked about, and found a significant difference between sPTB and TB (**Fig. 2a**).
- We performed a systematic analysis comparing the microdiversity of each phylogroup between term and preterm pregnancies. While we indeed found that most of the signal originated in significant differences in the microdiversity of several *Gardnerella* phylogroups, we also found an association between sPTB and the microdiversity of *Atopobium vaginae* (**Fig. S3**).
- We investigated the correlation between microdiversity and relative abundances systematically for all microbes, detecting a negative association for *L. iners* and *L. crispatus* (**Fig. S6**).
- Finally, we analyzed the association between antimicrobial resistance and prematurity at the metagenomic levels, for the entire vaginal microbiome.

Nevertheless, to completely address the reviewer's comment, we have included in the revision additional analyses on recombination (linkage disequilibrium r^2), maximal growth rate, and selection (dn/ds) for all phylogroup. We have found no statistically significant signal for these metrics in non-*Gardnerella* phylogroups. The results of these new analyses have been added to the manuscript (L297-298, 314-315, and 335-336).

4) The study would be much stronger if independent validation was carried out in another cohort. Is there a cohort from the public domain that could be utilized to confirm the presented findings?

To our knowledge, only a single other additional study performed metagenomic sequencing during pregnancy and collected pregnancy outcomes, that are needed for replicating this analysis - the 2018 study by Goltsman et al. Unfortunately, this study only profiled samples from 10 women, limiting the extent of validation that could be performed on it. **Notably, even with this small sample size, we have still detected increased microdiversity in the vaginal microbiome of women who delivered preterm, during the first half of their pregnancy ($P = 0.021$, Fig. 2h, i; included below for convenience), in line with the results of our study (Fig. 2f, g).** Since *Gardnerella* was only detected in 7 of these women (2 with preterm birth), we could not robustly validate our results regarding *Gardnerella*. We note, however, that among these seven participants, the two participants who delivered preterm had the highest and 4th highest *Gardnerella* microdiversity.

We thank the reviewer for this suggestion, resulting in this analysis which significantly strengthened the validity of our results and the strength of our claims.

Fig. 2 | Microdiversity patterns of the vaginal microbiome are associated with sPTB. f-g, A comparison of median genome-wide nucleotide diversity of all phylogroups between sPTB and TB, displayed for pregnancy S1 (d) and S2 (e). h-i, A comparison of median genome-wide nucleotide diversity of all phylogroups between sPTB and TB, displayed for pregnancy S1 (f) and S2 (g), using data from a cohort in Goltsman et al.

5) All the code to do the analysis should be made available.

We have published the code for the data analysis in a GitHub repository: https://github.com/korem-lab/MOMs-PI_microdiversity_2023.

Reviewer #2 (Remarks to the Author):

This is secondary analysis of existing metagenomic data sets from the MOMs study (175 women term and 40 sPTB with longitudinal sampling). It focuses on expanding on the previous analysis and micro-diversity. The genetic analysis of the *Gardnerella vaginalis* species/groups is fascinating and timely (given the burgeoning interest in the literature). AMR gene analysis is highly relevant although given we know that women with BV are at higher risk of PTB it is not surprising that the sPTB group a strong association between of AMR potential detected.

We appreciate the reviewer's recognition of the significance and timeliness of our analysis, including the AMR gene analysis.

While we agree that the association between AMR genes and pregnancy outcomes that we found in our study is perhaps intuitively not surprising due to potential links between BV and preterm birth, our revised manuscript now includes additional analyses showing that this association is not confounded by BV. As shown in the figure below, AMR gene count is not significantly different between women with and without BV for both halves of pregnancy ($P = 0.38$ and 0.5 , respectively), nor is it different between women with and without BV history ($P = 0.45$ and 0.1 , respectively). These results suggest that the association between AMR gene and pregnancy outcomes is independent of BV, which is a novel finding to our best knowledge. We have added these new analyses to the manuscript (see **Supplementary Fig. 8**; attached below for convenience; L379-387).

Supplementary Fig. 8 | Antimicrobial resistance (AMR) gene profiles of the vaginal microbiome are not associated with bacterial vaginosis (BV). a,b, Median count (along period) of AMR genes in the first (S1) and second (S2) halves of pregnancy, compared between women with and without BV (a) and between women with and without BV history (b). Box, IQR; line, median; whiskers, 1.5*IQR; p , two-sided Mann-Whitney U.

Comments

Please comment on why the sampling groups has a gestation gap with PTB being women delivering up to 37 weeks, but term women only selected from 39 weeks onwards. Also reflect on how this approach (and worries about overlap between late preterm and early term data) impacts on the clinical relevance of the information.

Our paper is relying upon an existing metagenomic dataset from the MOMs-PI study (Fettweis et al., 2019), which did not include term pregnancies between 37-38 weeks of gestation. As such, we followed the definition of sPTB and TB used in the first paper. As noted in Fettweis et al. (2019), the consideration for having a gap in the gestational weeks between sPTB and TB is to avoid complications potentially associated with early term birth (Leat et al., 2017; Murray et al., 2017; Boyle et al., 2012). We have added a statement about the reasoning behind the definition of sPTB and TB to the revised manuscript (L517-518). We thank the reviewer for pointing this out.

Comment on how findings in this refined analysis modifies conclusions drawn from previous analyses.

The associations between the taxonomic composition of the vaginal microbiome and preterm birth were consistent with previous analyses (see L128-132 and L424-425). We added to these previous studies by exploring an additional layer of microbial variability associated with sPTB - microbial genetic diversity - through population genetics analyses. The associations that we observed in nucleotide diversity, recombination, selection, and AMR gene profiles with preterm birth expand reveal additional, previously unknown processes occurring during pregnancy in addition to ecological processes: evolutionary processes and genetic adaptation, and are suggested to also play a role in pregnancy outcomes. We thank the reviewer for this comment, which we have incorporated into the manuscript in L423-435.

A slight issue that I had with the original paper, which may still be pertinent to the current paper is that there is no mention on how clinical management may have affected measurements of the vaginal microbiota, especially longitudinally. How many of the women received interventions for risk of preterm birth e.g. cervical suture, progesterone etc. These may have altered the microbiome composition across gestation (inflammatory responses change with intervention).

We thank the reviewer for this comment. To address it, we pulled information regarding cerclage placement and progesterone treatment for all women. In this dataset, only 7 women received a cerclage, while 149 women did not; similarly, only 8 women received progesterone, while 147 did not. To gauge if these procedures could confound pregnancy outcomes in this study, we checked their association, and found that pregnancy outcome is not dependent on cerclage nor progesterone (Fisher's exact $P = 1$ and $P = 0.25$, respectively). We also repeated the global microdiversity analysis performed in Fig. 2a only for women who received no cerclage and no progesterone. We still observed that it was significantly higher in sPTB in the first half of pregnancy (Mann Whitney U $P = 0.028$), and not in the second half ($P = 0.12$), as we have shown for all women (**Fig. 2f,g**). These results suggest that the association between the microdiversity of the vaginal microbiome and pregnancy outcome is not confounded by these interventions. We have added these new analyses and discussion to the manuscript (**Supplementary Fig. 4**; attached below for convenience; L209-216).

Supplementary Fig. 4 | The association between microdiversity and sPTB is not biased by medical interventions for risk of preterm birth, including cerclage and progesterone. a, b, A comparison of median genome-wide nucleotide diversity of *Gardnerella* spp. between sPTB and TB, , for women who did not receive cerclage nor progesterone during pregnancy, displayed for pregnancy S1 (a) and S2 (b). Box, IQR; line, median; whiskers, 1.5*IQR; p , two-sided Mann-Whitney.

Differences detected between the first half of pregnancy versus the second half of the pregnancy are reported – please can you confirm that they had samples from all the women in both time periods? Otherwise drop out (potentially linked to late miscarriage/early PTB could bias the data).

We thank the reviewer for this question. Indeed, having a bias between the groups in their distribution of samples could confound these analyses. As can be seen in the new **Supplementary Table 2**, there are no differences between the groups in terms of the total number of samples collected, the number of samples collected during the first or second halves of pregnancy, and in terms of the gestational age at the first and last samples. To answer the reviewer directly, as none of the women included in this study delivered at < 20 gestational weeks, there were samples from all women in both time periods. These indicate that dropping out due to late miscarriage/early PTB is not a concern to bias our findings in this study. We have added these new analyses and discussion to the method section in the manuscript (**Supplementary Fig. 9a**; attached below for convenience; L528-530) to address this concern.

Supplementary Table 2. Summary of longitudinal samples collected during pregnancy for women who deliver spontaneous preterm (sPTB) and at term (TB).

	sPTB	TB	Mann-Whitney U P
N of samples / woman (mean \pm s.d.)	3.36 \pm 1.51	3.21 \pm 1.13	0.83
N of samples / woman for the first half of pregnancy (mean \pm s.d.)	1.68 \pm 0.69	1.51 \pm 0.56	0.30
N of samples / woman for the second half of pregnancy (mean \pm s.d.)	2.34 \pm 1.10	2.14 \pm 0.90	0.27
Gestational age at the first sample (mean \pm s.d.)	17.38 \pm 8.08	16.09 \pm 7.50	0.45
Gestational age at the last sample (mean \pm s.d.)	31.23 \pm 5.52	32.31 \pm 3.66	0.72

Supplementary Fig. 9 | Distribution of longitudinal samples. a. Histogram showing the distribution of gestational age of women at delivery.

Also should suggest why there might be a difference in the discussion - is it that the hormonal milieu of pregnancy in the second half of pregnancy is beneficial (link in with other observations that the microbiome becomes more lactobacillus dominated in later pregnancy?)

The enriched nucleotide diversity during the first half of pregnancy might be related to the change of human chorionic gonadotropin (HCG), which peaks at roughly the same time as *G. vaginalis* microdiversity as we observed in Fig. 2c, and plays an immunomodulatory role in humans (Schumacher et al., 2009; Polese et al., 2014). We have added this hypothesis to the discussion section in the manuscript (L447-450).

Whilst in depth review of analytical approaches is outside of my expertise, they appear to be robust, appropriate, and well explained. It would be useful to know whether the code for any of the analysis is deposited for public access?

We have published the code for the data analysis in a GitHub repository: https://github.com/korem-lab/MOMs-PI_microdiversity_2023.

Propensity scores were used to assess a limited number of confounders, but as there are other confounders, particularly those that drive inflammation, can affect microbiome composition. It would be useful to discuss these and highlight in the limitations.

We agree that other unmeasured confounders (e.g., certain drugs or exogenous chemical compounds that may affect inflammation) may bias our findings. We have added a discussion highlighting this limitation in L431-435.

The authors in the methods state they account for false discovery rates. Is this undertaken for all the different types of analysis shown (e.g. Fig 4).

We thank the reviewer for this comment. We initially performed FDR in all cases where over 10 tests were performed. We have now added FDR values for Fig. 4g and 4h in L367 and L371, such that FDR correction is applied for all the different types of analysis shown.

Figure legends: It would be useful to give gestational time points studies and n numbers in the figure legends (e.g. Fig 1). I am aware some information provided in methods, but figure legends should give all information needed to interpret the data.

We thank the reviewer for pointing this out. We have added Ns to Fig. 1b and 1c. All figures now have the N numbers and gestational information specified.

Reviewer #3 (Remarks to the Author):

In this manuscript Lioa et. al. perform detailed metagenomic analyses on a set of previously sequenced stool samples from a cohort of infants born term and preterm. The main goal of the manuscript is to identify microbial signatures associated with preterm birth. The most substantial finding of the work is their identification of *Gardnerella* microdiversity being associated with preterm birth. Overall I found this manuscript to be technically sound, with some minor comments listed below, but a little difficult to interpret the biological underpinnings of the results. To me this study feels like a great exploratory look into the associations between microdiversity and preterm birth, and it provides solid justification for further research in this area.

We thank the reviewer for their appreciation of our study. We agree that this study serves to demonstrate the utility of population genetic analysis in this clinical and biological setting. We would like to point out that our analyses do go beyond these associations to try and understand the biological underpinning of our results. We present several complimentary pieces of evidence linking the increase in microdiversity to improved fitness of *Gardnerella* (**Fig. 3**). We offer that this increased fitness might relate to drug metabolism (**Fig. 2j**) and antimicrobial resistance (**Fig. 4**). As the reviewer mentions, we hope that these results motivate further research in these directions.

Specific comments:

1) A really great deal of work is put into showing that the association between *Gardnerella* and microdiversity is not due to sampling and sequencing bias, which I applaud the authors for, but I still have concerns that sequencing coverage may be impacting the result. On lines 261 to 271, the authors subset each sample to an identical number of reads, but if *Gardnerella* is more abundant in preterm birth, this will still result in higher coverage of *Gardnerella* in PTB samples. If coverage is associated with nucleotide diversity, this would explain the higher nucleotide diversity in PTB samples. A test that would evaluate this directly would be to subset a variable amount of reads from each sample to make *Gardnerella* have the same sequencing coverage / depth in all samples. This would directly control for any relationship between sequencing coverage and nucleotide diversity.

We thank the reviewer for this comment. We have performed the analysis as requested - we have mapped all reads to the *Gardnerella* phylogroups, and then randomly subsampled 5,000 *Gardnerella*-mapped reads from each sample, and repeated our analysis. Also in this analysis, the nucleotide diversity was higher in pregnancies that ended preterm compared to those ending at term (Mann Whitney U $P = 0.028$; now **Supplementary Fig. 5j** in the revised manuscript, also attached below; L276-280).

We would like to also point out that **Supplementary Fig. 5e** also shows that the number of reads mapped to a genome is not typically associated with its nucleotide diversity.

Supplementary Fig. 5 | The association between microdiversity and sPTB is not biased by sequencing depth. j. Median genome-wide nucleotide diversity along pregnancy of *Gardnerella* spp., compared between sPTB and TB based on 5,000 reads mapped to *Gardnerella* spp. sampled from each sample.

2) I like the analysis performed in lines 144 - 152, but it's unclear why *M. genomsp.* was the only phylogroup tested. Were all phylogroups tested and this was the only significant one? If so, was the proper FDR correction performed?

Yes, we tested all phylogroups and performed FDR correction. Only *M. genomsp.* was found to be significant. We have revised the text to make it clearer (L140).

3) In the analysis associated with Figure 3b, why did the authors choose to evaluate replication rates using codon bias instead of tools that profile *in situ* replication rates like iRep and PTR?

We indeed profiled *in situ* growth rate using both iRep and PTR. However, for iRep, the data was very sparse, preventing us from performing further analyses to compare between sPTB and TB. For PTR, we did not detect a significant difference. We now report this result in L299-300. As gRodon (the codon bias approach) estimates maximal growth rate instead of *in situ* growth rate, it could be that at the time the samples being sequenced, the association between the growth rate of *G. vaginalis* and preterm birth is weak, but the growth potential, regardless of environmental conditions, of *G. vaginalis*, is associated with preterm birth.

REVIEWERS' COMMENTS

Reviewer #1 (Remarks to the Author):

The authors have addressed my comments.

Reviewer #2 (Remarks to the Author):

The authors have answered all the questions posed in the original review thoroughly.

Reviewer #3 (Remarks to the Author):

All of my concerns have been addressed. I believe the paper is now ready for publication.

- Dr. Matt Olm